# Colorectal Cancer Liver Metastases: Genomics and Biomarkers with Focus on Local Therapies

**DOI:** 10.3390/cancers15061679

**Published:** 2023-03-09

**Authors:** Yuliya Kitsel, Timothy Cooke, Vlasios Sotirchos, Constantinos T. Sofocleous

**Affiliations:** 1Intervantional Oncology, IR Service, Department of Radiology, Memorial Sloan Kettering Cancer Center, New York, NY 10065, USA; 2College of Medicine, SUNY Downstate Health Sciences University, New York, NY 11203, USA

**Keywords:** colorectal cancer, liver metastases, biomarkers, genomics, interventional oncology

## Abstract

**Simple Summary:**

Colorectal cancer (CRC) is a leading cause of death among cancer patients, and the liver is the most common visceral metastatic site. Despite promising advances in treatment, recurrences with the eventual progression of disease or liver failure are common. Molecular cancer biomarkers are any measurable molecular indicator of the risk of cancer, the occurrence of cancer, or patient outcome, to help personalize treatment and to identify patients who may benefit most from a specific therapy. They may include germline or somatic genetic variants, epigenetic signatures, transcriptional changes, and proteomic signatures. The purpose of this work is to review the utility of tumor genomics and molecular biomarkers associated with colorectal cancer liver metastases (CRLM). As biomarkers have shown great potential for therapeutic decision making, this review would also be valuable for better understanding of CRLM treatment with a focus on local therapies and in particular image-guided liver-directed treatments.

**Abstract:**

Molecular cancer biomarkers help personalize treatment, predict oncologic outcomes, and identify patients who can benefit from specific targeted therapies. Colorectal cancer (CRC) is the third-most common cancer, with the liver being the most frequent visceral metastatic site. KRAS, NRAS, BRAF V600E Mutations, DNA Mismatch Repair Deficiency/Microsatellite Instability Status, HER2 Amplification, and NTRK Fusions are NCCN approved and actionable molecular biomarkers for colorectal cancer. Additional biomarkers are also described and can be helpful in different image-guided hepatic directed therapies specifically for CRLM. For example, tumors maintaining the Ki-67 proliferation marker after thermal ablation have been particularly resilient to ablation. Ablation margin was also shown to be an important factor in predicting local recurrence, with a ≥10 mm minimal ablation margin being required to attain local tumor control, especially for patients with mutant KRAS CRLM.

## 1. Introduction

Colorectal cancer (CRC) is the third-most common form of cancer with the fourth highest mortality rate worldwide [1]. Over half of CRC patients undergoing autopsy have been found to have liver metastases, suggesting that colorectal cancer liver metastases (CRLM) are a major cause of death in this population [2]. Despite promising advances in chemotherapy and liver-directed locoregional therapies (i.e., surgery, ablation, and arterially directed treatments), recurrences with eventual progression of disease and liver failure are common [3].

## 2. Genetic Alterations in Colorectal Cancer

Cancers are caused by the accumulation of mutations in genes that change the normal programming of cellular differentiation, proliferation, and death. Through the development of DNA sequencing, our understanding of the DNA changes associated with CRC is rapidly growing [4]. The ability to identify sequence variants in genes has improved our understanding of how cancer develops, allowing the development and application of treatments targeting these variants [4].

The Cancer Genome Atlas (TCGA) was created to study genomic changes in various cancers [4,5]. Each individual CRC contains between 60 and 1500 mutations. Few of these mutations have been found to be clinically relevant [6]. CRC generally divides biologically to those types exhibiting microsatellite instability (MSI) and those that are stable.

According to the TCGA, in non-hypermutated tumors (mutation rates of ≤12 per 10^6^ base pairs), the most commonly mutated genes were APC, TP53, KRAS, PIK3CA, NRAS, SMAD4, FBXW7, and TCF7L2 [7]. Mutated NRAS and KRAS genes typically displayed oncogenic mutations of codon 61 or codon 12 and 13 [7]. In hypermutated tumors, TGFBR2, APC, ACVR2A, MSH3, MSH6, TCF7L2, SLC9A9, and BRAF (V600E) were frequently mutated [7]. However, the APC and TP53 genes were more commonly mutated in non-hypermutated tumors versus hypermutated tumors: APC (81 vs. 51%) and TP53 (60 vs. 20%) [7]. These results suggest that hypermutant and non-hypermutant tumors advance through different series of genetic events.

Changes in MAPK, WNT, PI3K, TGF-b, and p53 signaling pathways are present in CRC [7]. WNT is the most prominent with regards to the carcinogenesis of colorectal cancer [8]. WNT signaling regulates the amount of β-catenin through processes that involve ubiquitin-mediated degradation and phosphorylation, thus impacting overall signal transduction [9]. Loss of adenomatous polyposis coli (APC), a negative regulator of the WNT pathway, is the hallmark of human CRC [9].

Different clinical and biologic characteristics have been observed between left-sided (arising from the splenic flexure, descending, or sigmoid colon) versus right-sided (arising from the cecum, ascending colon, hepatic flexure, or transverse colon) primary tumors. Genetic expression patterns differ between the right- and left-sided colonic epithelium, including a higher expression in cytochrome P-450 in the right colon, suggesting varying levels of exposure to ingested metabolites [10,11]. Overall, right-sided colon cancers carry a worse prognosis when compared to left-sided tumors and are more likely to have advanced TNM staging, peritoneal dissemination, MSI leading to a hypermutated state, BRAF/KRAS/PIK3Ca/SMAD4/FBXW7 mutations, and hypermethylation via the CpG island methylator phenotype (CIMP) [12,13,14]. On the other hand, mutations in TP53 and APC are often seen in left-sided CRC [15,16]. In addition to point mutations, left-sided CRC express more receptor tyrosine kinase amplifications, such as those in epidermal growth factor receptor (EGFR) and ERBB2 [10]. The variations in CRC sidedness can also have embryonic roots: the right side of the colon develops from the midgut, whereas the left side develops from the hindgut, influencing blood flow [17]. Lastly, gut microbiome differences can participate in the phenotypic expression of CRC sidedness [15].

Classifying tumor aggressiveness is based on several variables including the ability to create distant metastasis, lymph node status, tumor stage, and vascular invasion. Tumor aggressiveness has been associated with certain localized amplifications, deletions, and gene expression modification, including those of SCN5A, a well-known regulator of colon cancer [7]. Certain gene mutations also carry prognostic significance [18]. For example, highest cure rates have been observed in patients with NOTCH1 and PIK3C2B mutations, compared to patients with SMAD3 mutations, who had the lowest rates of cure [19].

## 3. Molecular Biomarkers of Colorectal Cancer

In the 1980s, the median overall survival (OS) of patients diagnosed with metastatic CRC (mCRC) did not exceed six months. The development and implementation of biomarkers, combined with significant advances in chemotherapeutics and locoregional treatments, have increased OS to a median of almost 30 months in the same population [20,21,22,23,24,25]. Elevated carcinoembryonic antigen (CEA) levels have been associated with poor prognosis and shorter OS following CRC resection. Of note, CEA levels can be elevated in other malignancies and inflammatory states, and therefore CEA is not a specific diagnostic biomarker [26]. Lack of normalization of CEA levels following resection is likely the result of inadequate resection and presence of metastatic disease. Routine CEA monitoring can help identify early, recurrent, or metastatic disease, that may benefit from additional therapies [27]. CEA is considered a sensitive indicator of post treatment recurrence [27]. Recent publications have indicated that preoperative CA19-9 is an independent prognostic factor in CRC patients, especially in the face of normal preoperative levels of serum CEA [28].

Currently, the National Comprehensive Cancer Network (NCCN) recommends testing for the following gene mutations in patients with CRC, as they are potentially actionable biomarkers that can guide therapeutic considerations: BRAF and KRAS/NRAS mutations, HER2 amplifications, and microsatellite instability high (MSI-H)/mismatch repair (MMR) (Table 1) [29].

### 3.1. KRAS and NRAS Mutations

RAS proteins are GTPases that regulate cell survival and proliferation. Human RAS genes, including Kirsten RAS (KRAS), Harvey RAS (HRAS), and neuroblastoma RAS (NRAS), are integral in GTPase activity [30]. Oncogenic mutations in RAS signaling pathways are associated with different aspects of cancer development. Since oncogenic mutations in various components of the RAS/MAPK pathway seem to be mutually exclusive in many tumors, deregulating RAS-dependent signaling is essential in tumorigenesis [31]. Regarding the prevalence of RAS mutations in CRC patients, a recent study showed that just over 50% of patients had KRAS mutations, with HRAS and NRAS mutations occurring at lower rates [32]. RAS signaling is not unique for CRC, as it is also present in non-malignant conditions such as diabetes, autoimmune, and inflammatory disorders [31].

Because the RAS/RAF/MEK/ERK pathway is downstream of EGFR, there are significant considerations that must be made regarding treatment of tumors expressing KRAS or NRAS mutations. Mutations within this pathway are robust negative predictive markers, and treatment with panitumumab or cetuximab is ineffective for tumors that have mutations in exons 2, 3, or 4 of the NRAS or KRAS genes [33,34,35,36,37,38]. Furthermore, these same mutations have been shown to express resistance to anti-EGFR therapy, both with and without chemotherapy, and correlate with worse outcomes [32,39,40,41,42]. Nevertheless, patients with left-sided CRC wild type RAS tumors have excellent outcomes when treated with anti-EGFR based therapy [43].

The NCCN Colon and Rectal Cancers Panel recommends determination of the RAS mutation status at diagnosis of stage IV disease [29]. Consistent with these recommendations, the American Society for Clinical Pathology (ASCP), Association for Molecular Pathology (AMP), College of American Pathologists (CAP), and the American Society of Clinical Oncology (ASCO) also developed a guideline on molecular biomarkers for CRC that also recommends RAS (KRAS/NRAS) genotyping of tumor tissue in all patients with mCRC [44]. KRAS and NRAS codons 12 and 13 (exon 2), 59 and 61 (exon 3), and 117 and 146 (exon 4) should all be included in mutational analysis [45,46]. Either the primary tumor or liver metastases can be tested for KRAS mutation analysis since there is a 96.4% concordance of mutational status between the primary sites and a metastasis [44]. Cetuximab or panitumumab are not indicated for the treatment of patients harboring NRAS or KRAS mutations, regardless of whether the drugs are used in isolation or with other anti-cancer therapies [47] since they will not respond to such agents. This minimizes toxicity and improves treatment cost-effectiveness. RAS genotyping of CRC at stage I, II, or I II is not recommended, and RAS testing should not be used for regimen selection in the first-line setting [44,48,49].

### 3.2. BRAF V600E Mutations

BRAF is a protein kinase downstream of RAS protein in the RAS-RAF-MEK-ERK kinase in the EGFR pathway [13,50]. RAF genes code for cytoplasmic serine/threonine protein kinase activity after binding RAS protein [51]. The RAS/RAF/MEK/MAP kinase pathway is an essential mechanism of tumor cell proliferation that mediates cellular responses to growth signals [52,53]. BRAF mutations, specifically with the V600E variant, occur in several malignancies [54]. In mCRC, BRAF mutations are found in 8–12% of patients (in 90% with V600E). BRAF-mutated tumors are more frequently seen in elderly and female patients [13,52,55,56,57]. Since BRAF is downstream of RAS, BRAF V600E mutations makes response to cetuximab or panitumumab highly unlikely, unless administered with a BRAF inhibitor [50,58,59].

At diagnosis of stage IV disease, NCCN recommends BRAF genotyping of either the primary or a metastatic tumor site as a predictive and prognostic marker for BRAF-targeted therapy [29,55,56]. BRAF V600E mutations have a poor prognosis regardless of treatment [55]. BRAF V600E mutation testing can be performed via direct DNA sequence analysis on formalin-fixed paraffin-embedded tissues and PCR amplification. Other methods for detecting this mutation include allele-specific polymerase chain reaction, immunohistochemistry (IHC), or next generation sequencing (NGS) [29].

A lower median OS for patients with BRAF-mutant mCRC compared to patients with wild-type BRAF (10.4 months vs. 34.7 months) has been reported [13]. Moreover, recurrences of resected stage III colon cancer had significantly worse survival in BRAF mutated tumors [60].

MSI-H/dMMR status is also strongly associated with the BRAF V600E mutation, with approximately 60% of MSI-H tumors having a BRAF mutation and only 5–10% of Microsatellite Stable (MSS) tumors having the same mutation in sporadic CRCs [61,62]. Because the BRAF V600E mutation is related to sporadic CRC patients but excludes Lynch syndrome, it is a key biomarker in distinguishing between the two etiologies [63].

In creating guidelines for CRC molecular biomarkers, the ASCP, CAP, AMP, and ASCO recommended that patients with dMMR tumors with loss of MLH1 receive BRAF p.V600 mutational analysis to evaluate for risk of Lynch syndrome. Although the absence of BRAF mutation does not completely exclude the risk of Lynch syndrome, the presence of a BRAF mutation would strongly favor a sporadic etiology [45].

### 3.3. Microsatellite Instability (MSI)/DNA Mismatch Repair Deficiency (dMMR) Status

Microsatellites are short segments of DNA that repeat at specific genomic locations. Under normal circumstances, in the event of an insertion or deletion within these regions, the MMR system rectifies any error. However, defects in this system cause deficient mismatch repair (dMMR). Although tumors with MSI retain their chromosomal number, they contain microsatellites that vary in length due to dMMR, which is thought to be a principle of early tumorigenesis [64]. About 15% of CRCs are affected by dMMR, which is more frequently seen in early stages of the disease (15–20% in stages I–II, 10–15% in stage III, and just 5% in stage IV) [65,66]. Up to 20% of patients with sporadic CRC have dMMR which is usually caused by hypermethylation of the MLH1 promoter [67,68,69].

Despite the numerous mutations present in dMMR tumors, mutant proteins are able to circumvent the immune system by binding to the T-effector cell’s programmed cell death protein (PD-1) receptor via programmed-death ligands 1 and 2 (PD-L1 and PD-L2) [70]. This system was initially thought to be protective for the host, but tumor cells that upregulate PD-1 have been shown to be elusive to the immune system [70]. As a result, it was theorized that dMMR tumors may have sensitivity to PD-1 inhibitors. This was validated in a small prospective phase 2 study, that showed complete response in 12 dMMR patients with stage II–III rectal cancer treated with single-agent dostarlimab (an anti-PD-1 monoclonal antibody), without any resection or radiotherapy [71]. There was no evidence of tumor in any patients after a minimum 6-month follow-up period [71]. In MSI-H CRCs, a recent study established a link between inflammatory states and suboptimal tumor response to PD-1 inhibition; an elevated neutrophil-to-lymphocyte ratio indicated poor tumor response to checkpoint inhibitors [72].

As a result, the NCCN now recommends MSI or MMR testing for any patient with a history of CRC, and checkpoint inhibitors for dMMR/MSI-H disease [29]. dMMR/MSI-H testing is essential for four main reasons. Firstly, because of the potential connection between dMMR and Lynch syndrome, it is a potent screening test to identify the most common cause of hereditary CRC [73]. Secondly, identifying dMMR status has proven to be one of the best ways to improve patient prognosis, particularly in those with stage II CRC, where MSI-H was associated with longer OS, relapse-free survival, and time to recurrence when compared to MSS [74,75]. Moreover, dMMR status was a strong negative predictor of 5-fluorouracil efficacy, making it a good marker to guide chemotherapy choice [76]. Finally, MSI-H/dMMR status increased the ability to predict response to checkpoint inhibitors [77,78]. Patients previously treated for MSI-H/dMMR mCRC experienced a 30% plateau in OS and progression-free survival (PFS) at five years in the phase II KEYNOTE-164 study [26,79].

dMMR status can be tested via IHC, and MSI status can be tested using PCR or NGS based assays. IHC is usually the test of choice because of the test’s high concordance (>90%) [80], but NGS or PCR testing can be used if IHC findings are inconclusive [81,82]. There is no need to analyze both the primary and the metastatic sites since concordance is over 90% [82].

### 3.4. HER2 Amplification

Human epidermal growth factor receptor 2 (HER2) is a factor involved in multiple tumor types, including colorectal, breast, and gastroesophageal cancers [83]. HER2 (also known as ERBB2) codes for a transmembrane protein on the long arm of chromosome 17 and activates downstream signaling pathways by complexes with other HER proteins [84]. Cell differentiation, migration, proliferation, and apoptosis inhibition are all facilitated by HER2 [85]. IHC and fluorescence in situ (FISH) can be used to identify HER2 overexpression and amplification [86].

HER2 amplification is present in 2–5% of all CRCs [87] with higher prevalence in wild type RAS/BRAF tumors (approximately 5–14%) [88,89]. Following anti-EGFR treatments, the frequency of HER2 overexpression increases, possibly suggesting HER2′s involvement in resistance to the anti-EGFR agents [90,91].

The NCCN Guidelines recommend HER2 amplification testing in mCRC patients, but not for KRAS/NRAS or BRAF mutant tumors [29]. It is shown that in patients with wild-type RAS/BRAF mCRC, the HER2 status did not impact median PFS on therapy without an EGFR inhibitor [92]. Thus, in patients with wild-type RAS and BRAF tumors with HER2 overexpression, HER2 targeted treatment can be used as a subsequent therapy [88,93]. Additionally, previously treated patients with HER2-positive mCRC responded to trastuzumab deruxtecan, an antibody-drug conjugate, with a 45% response rate [94]. Although HER2 is a predictive biomarker of resistance to monoclonal antibodies that target EGFR [89,92,95] the current level of evidence does not support HER2 overexpression as a prognostic tool [96].

### 3.5. NTRK Fusions

Neurotrophic Tropomyosin Receptor Kinase (NTRK) genes encode the tropomyosin receptor kinase (TRK) proteins involved in cell homeostasis and embryonal neural development [97,98]. These genes can fuse with other genes, producing NTRK gene fusions that can lead to uncontrolled cell growth and division [99]. These can be detected with IHC, plasma cell-free DNA profiling, and tumor DNA and RNA sequencing [100]. NTRK gene fusions exist in several cancers, including CRC, where they are present in only <1% [101,102]. NTRK fusions are more common in MSI-H/dMMR CRC [103] and are connected to APC and TP53 mutations [104]. Patients with NTRK gene fusion-positive mCRC are good candidates for TRK inhibitors therapies. According to the NCCN guidelines; larotrectinib and entrectinib have received FDA approval for treating patients with NTRK gene fusion-positive metastatic, unresectable solid tumors [29].

## 4. Biomarkers in the Surgical Treatment of CRLM

Despite advances in surgery, complete CRLM resection is feasible in a select subgroup of patients. Surgical cohorts have shown that resection of CRLM is an effective therapeutic approach [105,106] that can result in a prolonged disease-free state and subsequent OS of 10 years or more in low-risk patients treated with clear resection margins (R0) [107,108,109]. There is an R classification to indicate the presence of residual tumor post-resection. In this classification, R0 = no residual tumor, R1 = microscopic residual tumor, and R2 = gross residual tumor [110]. R0 and a 3-year recurrence free-survival after hepatectomy of CRLM were factors that predicted the probability for long-term survival [108,111].

Patients are still at risk of recurrence especially within 2 years after resection [108]. Randomized controlled trials (RCTs) typically report five-year OS, but the evidence does not support the notion that resection is curative. Following CRLM resection, the 5- and 10-year survival was 50% and 25%, respectively [112,113,114].

Long-term outcomes of hepatic resection can be predicted using the clinical risk score (CRS). The factors evaluated in the scoring system are (1) node-positive primary, (2) the number of hepatic tumors (>1), (3) the largest hepatic tumor size >5 cm, (4) the disease-free interval from the detection of the primary to the development of metastases < 12 months, and (5) CEA level > 200 ng/mL. A total score is calculated by allocating one point to each variable. Patients with up to two positive criteria were found to have favorable OS after surgery [115].

BRAF, KRAS, and histological growth pattern (HGP) further enhance post-resection prognosis and can predict 5-year OS [116,117]. KRAS mutant patients had worse OS compared to wild-type RAS patients after repeat hepatectomy for CRLM (median OS 42.5 vs. 26.6 months) [118]. Lung recurrences are more common in patients with KRAS mutations [119,120,121]. Moreover, KRAS mutant patients with resected CRLM treated with adjuvant hepatic arterial infusion and systemic chemotherapy had worse 3-year resection-free survival and increased incidence of bone, brain, and lung metastases compared to KRAS wild-type patients [122].

Positive resection margins were more common in patients with mutant KRAS tumors (11.4% vs. 5.4%) [123]. Interestingly, a study showed that a R0 resection margin provided a survival benefit to patients with wild-type KRAS tumors, in contrast to those with KRAS mutant tumors after stratification based on margins [124]. Moreover, even a 1 cm margin did not improve OS in patients with KRAS mutant tumors [125]. On the other hand, non-anatomical resections were associated with worse disease-free survival in patients with KRAS mutated tumors; therefore, more extensive anatomical resections may be required in this population [126].

Patients undergoing CRLM resection with BRAF mutation have worse prognosis [127,128]. When patients were categorized according to synchronous/metachronous CRLM presentation, BRAF/KRAS mutations were revealed to be independent prognostic biomarkers among patients with exclusively synchronous CRLM [128]. In the first postoperative year, BRAF mutational status was the most important prognostic variable connected to survival over time. After the first year, resected extrahepatic disease or R1 margin was the principal predictor of prognosis [19,128].

Circulating tumor DNA (ctDNA) is a promising novel marker that can be used for detecting postsurgical minimal residual disease. Postoperative ctDNA-positive patients had significantly shorter relapse-free survival (RFS) compared to ctDNA-negative patients (12.7 vs. 27.4 months) [129]. A recent study reported that, following CRLM resection, patients with ctDNA+ were twice as likely to have a co-mutation involving RAS+ and TP53, that is considered a significant prognostic biomarker (47% vs. 23%) [130]. Within one year of hepatectomy, recurrence rates were significantly higher in ctDNA+ compared to ctDNA- patients (94% vs. 49%) [130].

## 5. Biomarkers and Interventional Oncology for CRLM

### 5.1. Thermal Ablation (TA)

Thermal ablation (TA) is a local curative therapy for CRLM, either as a monotherapy or in conjunction with surgery, in select patients with small volume disease and where all visible disease can be eliminated [29]. TA is an efficacious and safe therapy that can offer local tumor control and improved OS when the tumors are treated with sufficient margins [131,132,133]. TA can be used as an alternative to surgical treatment or in conjunction with resection, as well as a salvage therapy for liver tumor recurrences after resection [134,135]. The main objective is to achieve tumor necrosis with adequate margins, while maximally sparing healthy surrounding liver parenchyma [136]. Image-guided TA is commonly used in patients who cannot undergo or decline surgery with limited tumors (up to 3) and small tumor size (ideally under 3 cm) [137]. TA can also be offered in surgical candidates with the concept of the “test-of time” approach that could potentially spare patients with aggressive disease biology from morbid liver resection [138].

Ablation outcomes, particularly patient survival (including patients that were not surgical candidates), are comparable to resection [139]. The results of percutaneous ablation have been enhanced by recent software applications that assist in intra-procedural response assessment and planning [140,141,142,143,144].

The clinical risk score for surgical resection mentioned above was modified and validated as a predictor of local tumor progression-free survival (LTPFS) in patients undergoing radiofrequency ablation of CRLM. The two modifications were (1) decreasing the tumor size cutoff from 5 cm to 3 cm and (2) decreasing the CEA level cutoff from 200 ng/mL to 30 ng/mL [134,145].

Minimal ablation margin size and tumor size are two of the most critical surrogate imaging markers and technical endpoints influencing ablation success and long-term local tumor control [140,145,146,147,148].

A robust independent predictor of local tumor progression (LTP) and OS after CRLM ablation is the detection of tumor cells expressing Ki-67 in ablation zone biopsy specimens, suggesting that these tumors may harbor ablation resistance mechanisms [149,150,151,152,153]. Ki-67 positive tumor cells on the electrode after radiofrequency ablation (RFA) of liver tumors is an independent predictor of OS and LTPFS [150,151].

Although size was thought to be an independent risk factor for LTP in tumors 3–5 cm, it is not as significant of a variable when the data are stratified by margins [151]. Subsequent prospective trials indicated that the combination of a minimal margin of >5 mm and a negative for tumor biopsy of the ablation zone center and margin offer consistent local tumor control over 97% after RFA [149] and over 93% for RFA and microwave ablation (MWA) [152]. In addition, the use of immediate fluorescent stains on ablation zone tissue can detect necrotic versus residual viable tumor intra-procedurally and can predict LTP at 12 months [153]. Similarly, real-time metabolic imaging has been used as a surrogate imaging biomarker of residual disease after ablation and is predictive of LTP [154,155,156,157,158,159].

Numerous studies found that patients undergoing percutaneous image-guided ablation were at higher risk for LTP if they had mutant RAS CRLM [148,160,161]. Patients with mutant RAS also had a lower 3-year LTPFS compared to those with wild-type RAS [160]. In the mutant RAS patient population, ablation margin has proven to be one of the most important factors in improving LTPFS [148]. Mutant RAS CRLM with minimal ablation margins of both 1–5 mm and ≥10 mm had significantly higher LTP rates when compared to wild-type RAS CRLM with similar margins [145,148,160,161]. A minimal ablation margin of >10 mm can improve LTPFS in mutant RAS CRLM [148] and should be considered a critical technical endpoint to achieve local tumor control in these patients [145,148].

### 5.2. Transarterial Radioembolization

Transarterial radioembolization (TARE) is a therapy that delivers yttrium-90 (Y90) microspheres, a beta-emitting radionuclide, intraarterially into liver tumors via the hepatic artery [162]. TARE mechanisms of action are multi-faceted and include causing tumor necrosis and fibrosis, impacting tumor vasculature and dystrophic calcification resulting in a significant prolongation of the median time to hepatic tumor progression when administered early in the disease course with first-line chemotherapy [163,164,165]. TARE is an effective locoregional therapy available to patients with primary and metastatic liver malignancies including unresectable and/or chemorefractory colorectal cancer hepatic disease [166]. As the clinical use of TARE increases in the era of personalized cancer care, the ability to predict tumor response is an essential tool in the clinical decision making to improve treatment outcomes [166].

Traditional biomarkers such as CEA and AST can predict oncologic outcomes after TARE of CRLM [167]. At the time of TARE, patients with low CEA (<20 ng/mL) had a better OS compared to those with high CEA (>20 ng/mL) (15.9 months vs. 9.6 months) (167). Furthermore, patients with low AST (<40 IU) had a better OS compared to those with a high AST (>40 IU) (13.2 months vs. 7.9 months) [167]. A predictive nomogram including the CEA, ALT, tumor size and biology, the presence of extrahepatic disease, and albumin level was highly predictive of 12-month patient survival after TARE of CRLM in heavily pretreated patients in the chemorefractory stage [168].

With regards to oncogenic biomarkers, mutant KRAS, MAPK, and PI3K genes have all been shown to independently predict decreased OS in patients undergoing TARE [169,170,171]. A major predictive genetic biomarker is wild-type vs. mutant KRAS tumor status. Patients with wild-type KRAS tumor status had significantly increased OS compared to those with mutant KRAS tumor status (9.5 months vs. 4.8 months) [169]. Additionally, mutant KRAS tumor status was shown to be an independent prognostic factor for PFS (91 days vs. 166 days, RAS-mutant vs. RAS-wild-type) [172]. Another oncogenic molecular pathway that has shown to play a role in TARE outcomes is MAPK. MAPK was only shown to affect OS in patients who progressed after multiple lines of chemotherapy prior to TARE. Patients with wild-type MAPK who received TARE after multiples lines of chemotherapy showed improved OS when compared to mutant MAPK (median OS 29.9 months vs. 21.3 months). This statistical significance was absent in the cohort of patients treated for disease present after the first line of chemotherapy [170]. Overall, there was a 33% and 55% incidence of progression at 6 and 12 months with mutant PI3K and 76% and 92% in patients with wild-type PI3K, indicating that the presence of an activating mutation in the PI3K pathway was linked to a longer overall time to progression [171].

Another consistent prognostic biomarker is the neutrophil–lymphocyte ratio (NLR). Patients with high NLR (over 4.6) had a lower OS compared to patients with a normal/low NLR (5.6 months vs. 10.6 months) [173]. Although it is not clear why a high NLR yields a lower OS, it is hypothesized that increased angiogenic factors and decreased lymphocytic response provide the tumor cells with the potential to resist and survive after treatment [166]. Higher nucleosomes measurements were obtained 24 h after TARE in patients suffering from disease progression compared to those without disease progression, presumably because of the increasing number of dysfunctional cells [174]. High pre-therapeutic and 24-h post-treatment values in serum high mobility group box 1 (HMGB1), a nucleoprotein inflammatory cytokine that triggers protective T-cell response and promotes tumor neoangiogenesis, were associated with lower OS [175].

In recent years, 18F-FLT (fluorotimidin) PET/CT has been discovered as a potential new prognostic imaging biomarker for TARE. A longer PFS with at least 30% decrease in 18F-FLT SUVmax and SUVpeak (*p* < 0.009) was noted in the treated hepatic lobe [176].

Metabolic tumor volume (MTV) and total lesion glycolysis (TLG) on FDG-PET/CT were found to be predictors of OS after TARE of CLM in comparison with SUVmax, SUVpeak, and RECIST 1.0 [177]. Additionally, tumor attenuation criteria, Choi criteria, and EORTC PET criteria predict liver PFS [178].

### 5.3. Transarterial Chemoembolization (TACE)

TACE is a catheter-based liver treatment that delivers chemotherapy and embolic material trans-arterially via the hepatic artery. This therapy causes tumor ischemia and necrosis by interrupting blood supply and locally delivering high doses of chemotherapy while reducing systemic toxicity. The embolic and chemotherapeutic mechanisms are synergistic, as the anti-tumoral agents can act on malignant tissue for longer periods of time [179].

TACE can also be used with irinotecan-loaded drug eluting beads, otherwise known as DEBIRI. While embolizing specific parts of the hepatic artery, DEBIRI utilizes irinotecan which rapidly releases from the drug eluting beads, thus increasing local concentration and reducing systemic exposure to the drug [180,181]. Despite numerous studies on DEBIRI showing promising outcomes, traditionally prognostic biomarkers such as CEA and CA 19-9 have not yielded statistical significance. In a study of 40 patients, 17 patients showed a ≥20% decrease in CEA levels after the first DEBIRI course corresponding to a significantly longer OS compared to other patients (9 vs. 5 months) [182]. Similar results were seen in CA 19-9 [182]. One biomarker, vascular endothelial growth factor receptor 1 (VEGFR1), did show a significant decrease 24 h after receiving DEBIRI, presumably due to the anti-tumor effect of anti-VEGF therapies [183]. Interestingly, VEGF and VEGFR2 did not significantly predict treatment efficacy or risk of adverse events [183].

## 6. Tumor Biomarkers and Hepatic Arterial Infusion Pump for CRLM

Hepatic artery infusion pump (HAIP) is a therapy that selectively delivers chemotherapy to the tumor and utilizes the first-pass effect of the liver, allowing for locally directed administration of high dose anti-cancer medication with minimal systemic exposure and tissue sparing [184]. In North America, the most commonly used chemotherapeutic agent in HAIP is floxuridine [185]. Additionally, HAIP floxuridine achieves very high response rates in the liver when combined with systemic therapy [184,186,187]. In one study, systemic chemotherapy and adjuvant HAIP therapy were administered to 169 patients with resected CRLM. 118 of these patients had wild-type KRAS tumors and 51 had tumors with KRAS mutations. Patients with wild-type KRAS tumors had a 3-year RFS of 46%, while those with mutant KRAS tumors had a 3-year RFS of 30%, and the 3-year OS was 95% and 81%, respectively. On multivariate analysis, KRAS independently predicted RFS (HR 1.9). These findings suggest that KRAS mutation is linked to an aggressive biology and a poorer prognosis following CRLM resection [122]. Cell free DNA (cfDNA) has the potential for prediction and prognosis of patients with CRLM receiving HAIP. Patients with a cfDNA value above the 75th percentile had a poorer median overall survival than patients below the 75th quartile (2.4 years vs. 3.9 years). In comparison to non-responders, baseline cfDNA levels were considerably lower in patients who had an objective response (0.91 ng/µL vs. 1.79 ng/µL) [188]. Furthermore, with an HR of 1.01 in multivariate analysis, increased CEA fails to demonstrate an association with death. This makes cfDNA a stronger predictive marker for mortality than CEA [189].

## 7. Conclusions

In conclusion, molecular profiling has become part of the mainstay in the management of the patient with colorectal cancer from early to advance stages of disease. It provides important prognostic information that can be used for patient counseling. More importantly, it is used in the therapeutic decision-making process and has implications on local therapies such as liver resection, ablation, or intraarterial therapies for metastatic liver disease. In several of these therapies, molecular profiling can also guide the extent of treatment as reflected in resection or ablation margins and RAS status. With the continuous evolution of DNA and molecular analysis techniques, genetic profiling will increase to affect personalized cancer therapies and help improve oncologic outcomes. A summary of the clinical implications of relevant biomarkers for liver treatments is presented in Table 2.

## Figures and Tables

**Table 1 cancers-15-01679-t001:** NCCN recommended biomarkers for CRC.

Biomarkers	Clinical Implication
KRAS/NRAS	Anti-EGFR therapy are not indicated for the treatment of patients harboring NRAS or KRAS mutations.
BRAF V600E	Poor prognosis regardless of treatment. Prognostic marker for BRAF-targeted therapy.
	Related to sporadic CRC, excludes Lynch syndrome.
MSI/dMMR	Improve patient prognosis. dMMR’s ties to Lynch syndrome. Prognostic marker for checkpoint inhibitors therapy. dMMR status is a strong negative predictor of 5-fluorouracil efficacy.
HER2	Predictive biomarker for HER2-targeted therapy in patients with wild-type RAS and BRAF tumors. Predict resistance to monoclonal antibodies that target EGFR.

CRC: colorectal cancer, HER2: human epidermal growth factor receptor 2, MSI: Microsatellite Instability, dMMR: DNA Mismatch Repair Deficiency, EGFR: endothelial growth factor receptor.

**Table 2 cancers-15-01679-t002:** Clinical implications of relevant biomarkers with regards to liver directed therapies.

Intervention	Biomarker	Clinical Implication
Hepatectomy		
Resection Margin	MUT KRAS	Reduced median OS, especially with repeat hepatectomy. Higher incidence of positive surgical resection margins.
	MUT BRAF	Strongest prognostic indicator in the first postoperative year for synchronous CRLM presentation.
	ctDNA+	Commonly co-mutated with RAS/TP53. Associated with early recurrence 12 months or earlier post-hepatectomy.
Interventional Oncology		
Ablation	Ki67+	Strong predictor of LTP and OS post-ablation. Positivity suggests ablation resistant mechanisms.
	MUT RAS	Higher risk of LTP and lower 3-year LTPFS.
	Margin	Minimum ablation margin of ≥10 mm can optimize local tumor control.
TARE	CEA (>20 ng/mL)	Significantly decreased OS.
	AST (>40 IU)	Significantly decreased OS.
	MUT KRAS	Significantly decreased OS. Independent prognostic factor of PFS.
	MUT MAPK	Reduced OS in patients with multiple failed lines of chemotherapy pre-TARE.
	MUT PI3K	Associated with longer time to local progression post-TARE.
	18F-FLT, 18F-FDG	Longer PFS in target liver lobe in patients with a 30% decrease in 18F-FLT. MTV and TLG as predictors of OS after TARE
TACE (DEBIRI)	CEA (>20 ng/mL)	Decreases in these biomarkers have no significant correlation with tumor radiological response.
	CA 19-9	
	VEGFR1	Sharp decreases were seen 24 h post-DEBIRI.
HAIP	MUT KRAS	Independent predictor of reduced RFS and 3-year OS.
	cfDNA	Over 75th percentile associated with poorer OS. Strong predictor of mortality.

MUT: mutant, OS: overall survival, CRLM: colorectal cancer liver metastases, LTP: local tumor progression, LTPFS: local tumor progression-free survival, PFS: progression-free survival, TARE: transarterial radioembolization, TACE: transarterial chemoembolization, DEBIRI: irinotecan-loaded drug-eluting beads, HAIP: hepatic artery infusion pump, ctDNA: circulating tumor DNA, CEA: carcinoembryonic antigen, AST: aspartate transaminase, MAPK: mitogen-activated protein kinase, PI3K: phophatidylinositol 3-kinase, FLT: fluorothymidine, CA: cancer antigen, VEGFR: vascular endothelial growth factor receptor, cfDNA: cell-free DNA, MTV: metabolic tumor volume, TLG: total lesion glycolysis.

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
