# Peer review of "Colorectal Cancer Liver Metastases: Genomics and Biomarkers with Focus on Local Therapies"

_cancers, 2023, doi:10.3390/cancers15061679_

Round 1

Reviewer 1 Report

In this manuscript, Yuliya Kitsel and coauthors reviewed genetic or molecular biomarkers for colorectal cancer liver metastases (CRLM), especially focusing on local therapies such as hepatectomy, ablation, and intraarterial therapies. As biomarkers have shown great potential for therapeutic decision-making, this review would be valuable for better understanding in the field of CRLM treatment. Below, I point out a few of comments that may improve current version of manuscript.

For the hotspot mutation analysis of the KRAS and BRAF gene, advanced technologies called liquid biopsy and gene panel test are getting increased attention. How’s about adding more discussion of advanced tools in the analysis of biomarkers?

Page 2, lane 36; instead of the subtitle “Genomes of colorectal cancer”, it would be better to say “Genetic alterations in colorectal cancer”.

Page 2, lane 47; it would be necessary to correct for the part of mutation rates, 106 base pairs à 106 base pairs.

Page 2, lane 50; regarding oncogenic mutations of NRAS/KRAS, not codon 16, but codon 61 is correct.

Author Response

Dear Reviewer, 

Thank you so much for your priceless noticing of small mistakes, which is very important! I corrected it (rows 36, 47 and 50). As this paper addresses mostly for clinical use, we did not go so deep in new exciting technologies in mutation analysis. But it is a great topic for other review paper!

Thank you again for your time and work on this paper!

With kind regards,

Yuliya

Reviewer 2 Report

This review summarizes current information on genomics and relevant biomarkers in Colorectal Cancer Liver Metastases. However, the authors should supplement and correct a number of sections of the report to make it better.

1. the review contains sufficient information on markers in metastatic colorectal cancer but an insufficient author's discussion. The material is now more descriptive than analytical

2. The material lacks data on a number of significant gene alterations: Myc, kit, etc.

3. insufficient attention is paid to tumour-infiltrating lymphocytes and immune checkpoints.

4. The review does not mention the importance of ctDNA for tumourigenesis.

Author Response

Dear Reviewer, 

Thank you so much for your comments and interest in this paper. 

  1. The discussion was limited in the general part of this review. And we purposely expand discussions in the Liver directed therapy part of the manuscript that is the main focus of this review.
  2. As the main focus of our paper is the clinical application of genomics and relevant biomarkers in Colorectal Cancer Liver Metastases, we described mostly gene alterations which are mentioned in clinical guidelines, like NCCN and others.
  3. Tumor-infiltrating lymphocytes and immune checkpoints are big and very interesting topics from immunology part of oncology. It deserves a separate review paper.
  4. For this comment the same reply as for comment 2.

Thank you again for your time and work on this paper! And for interesting ideas for our next review papers!

With kind regards,

Yuliya

Reviewer 3 Report

The review of Kitsel et al ” Colorectal Cancer Liver Metastases: Genomics and Biomarkers 2 with focus on local therapies.” is a well written manuscript and deserves publication.

I have just a few comments:

  • please write gene names in Italic
  • row 50: codon 61, not 16
  • it is not necessary to include full gene names in table notes
  • row 119: RAS pathway is not "defective", since RAS mutations lead to gain of function
  • row 144: "I, II, or II" should be "I, II, or III"
  • row 227: Since FISH detects HER2 amplification, I would say "IHC and FISH can be used to identify HER2 overexpression and amplification"
  • rows 232-233: please, rephrase. Ref 29 says the if the tumor has KRAS/NRAS/BRAF mutation, HER2 testing is not indicated

Author Response

Dear Reviewer, 

Thank you so much for your priceless noticing of small mistakes, which is very important! I corrected it (rows 50, 119, 144, 227 and 232-233), also I excluded full gene names in table notes. As it is not mandatory to write gene names in Italic, we will skip this moment. 

Thank you again for your time and work on this paper!

With kind regards,
Yuliya

Round 2

Reviewer 2 Report

Thank You for answer.